# 13-Acetoxysarcocrassolide Exhibits Cytotoxic Activity against Oral Cancer Cells through the Interruption of the Keap1/Nrf2/p62/SQSTM1 Pathway: The Need to Move Beyond Classical Concepts

**DOI:** 10.3390/md18080382

**Published:** 2020-07-23

**Authors:** Yi-Chang Liu, Bo-Rong Peng, Kai-Cheng Hsu, Mohamed El-Shazly, Shou-Ping Shih, Tony Eight Lin, Fu-Wen Kuo, Yi-Cheng Chou, Hung-Yu Lin, Mei-Chin Lu

**Affiliations:** 1Division of Hematology-Oncology, Department of Internal Medicine, Kaohsiung Medical University Hospital, Kaohsiung 807, Taiwan; ycliu@cc.kmu.edu.tw; 2Department of Internal Medicine, Faculty of Medicine, College of Medicine, Kaohsiung Medical University, Kaohsiung 807, Taiwan; 3Doctoral Degree Program in Marine Biotechnology, National Sun Yat-Sen University (NSYSU), 70 Lien-Hai Road, Kaohsiung 80424, Taiwan; pengpojung@gmail.com (B.-R.P.); m6430005@hotmail.com (S.-P.S.); 4Doctoral Degree Program in Marine Biotechnology, Academia Sinica, 128 Academia Road, Section 2, Nankang, Taipei 11529, Taiwan; 5Graduate Institute of Cancer Biology and Drug Discovery, College of Medical Science and Technology, Taipei Medical University, Taipei 106, Taiwan; piki@tmu.edu.tw; 6Ph.D. Program for Cancer Molecular Biology and Drug Discovery, College of Medical Science and Technology, Taipei Medical University, Taipei 106, Taiwan; tonyelin@tmu.edu.tw; 7Biomedical Commercialization Center, Taipei Medical University, Taipei 106, Taiwan; 8Department of Pharmacognosy, Faculty of Pharmacy, Ain-Shams University, Organization of African Unity Street, Abassia, Cairo 11566, Egypt; mohamed.elshazly@pharma.asu.edu.eg; 9Department of Pharmaceutical Biology, Faculty of Pharmacy and Biotechnology, German University in Cairo, Cairo 11835, Egypt; 10Graduate Institute of Marine Biology, National Dong Hwa University, Pingtung 944, Taiwan; fuwen@nmmba.gov.tw (F.-W.K.); judy811127q@gmail.com (Y.-C.C.); 11National Museum of Marine Biology & Aquarium, Pingtung 944, Taiwan; 12Division of Urology, Department of Surgery, School of Medicine, College of Medicine, I-SHOU University, E-Da Cancer & E-Da Hospital, Kaohsiung 824, Taiwan

**Keywords:** apoptosis, reactive oxygen species, 13-Acetoxysarcocrassolide, oral cancer, oxidative stress, Keap1/Nrf2/p62/SQSTM1

## Abstract

13-Acetoxysarcocrassolide (13-AC), a marine cytotoxic product isolated from the alcyonacean coral *Lobophytum crassum*, exhibited potent antitumor and immunostimulant effects as reported in previous studies. However, the 13-AC antitumor mechanism of action against oral cancer cells remains unclear. The activity of 13-AC against Ca9-22 cancer cells was determined using MTT assay, flow cytometric analysis, immunofluorescence, immunoprecipitation, Western blotting, and siRNA. 13-AC induced apoptosis in oral cancer cells Ca9-22 through the disruption of mitochondrial membrane potential (MMP) and the stimulation of reactive oxygen species (ROS) generation. It increased the expression of apoptosis- and DNA damage-related proteins in a concentration- and time-dependent manner. It exerted potent antitumor effect against oral cancer cells, as demonstrated by the in vivo xenograft animal model. It significantly reduced the tumor volume (55.29%) and tumor weight (90.33%). The pretreatment of Ca9-22 cells with N-acetylcysteine (NAC) inhibited ROS production resulting in the attenuation of the cytotoxic activity of 13-AC. The induction of the Keap1-Nrf2 pathway and the promotion of p62/SQSTM1 were observed in Ca9-22 cells treated with 13-AC. The knockdown of p62 expression by siRNA transfection significantly attenuated the effect of 13-AC on the inhibition of cell viability. Our results indicate that 13-AC exerted its cytotoxic activity through the promotion of ROS generation and the suppression of the antioxidant enzyme activity. The apoptotic effect of 13-AC was found to be mediated through the interruption of the Keap1/Nrf2/p62/SQSTM1 pathway, suggesting its potential future application as an anticancer agent.

## 1. Introduction

Head and neck cancers represent a heterogeneous disease group in which 90% of these cancers are squamous cell carcinoma. The treatment strategy of these cancers remains challenging and a multidisciplinary approach is necessary. High dose cisplatin accompanied by radiotherapy is currently the standard protocol for the primary treatment or postoperative therapy [1]. The combined use of chemo- and radiotherapy can induce high rates of acute and long-term toxicities, which may compromise the treatment response and patient’s quality of life. Local recurrence or distant metastasis may occur during the follow-up stages. Clinicians use anti-epidermal growth factor receptor (EGFR) therapy and immune checkpoint inhibitors along with traditional cytotoxic chemotherapy in patients with recurrent or metastatic cancer [2]. However, immunotherapy-related adverse effects remain a critical issue [3], and the identification of reliable, predictive, and prognostic markers needs more validation by further clinical trials [4]. Poor prognosis is usually observed in patients with recurrent or metastatic disease. Thus, there is an eminent medical need to explore novel therapy for head and neck cancers.

Nuclear factor erythroid 2-related factor 2 (Nrf2) is the master regulator of the cellular antioxidant response, a target of chemopreventive compounds, and a driver of cancer progression [5]. Recent studies demonstrated the critical role of Nrf2 in promoting cell survival and drug resistance in cells harboring oncogenic K-ras. The inhibition of Nrf2 can be an attractive strategy to increase the therapeutic effect of anticancer drugs and overcome drug resistance in cancers with oncogenic K-ras activation [6]. The inhibition of Nrf2 can improve artesunate induced ferroptosis in cisplatin-resistant head and neck cancers [7]. Compounds that inhibit Nrf2 can sensitize cancer cells to chemotherapeutic agents, and it is expected that these compounds will be tested in clinical trials [8]. Finding new and effective inhibitors of Nrf2 is an attractive goal to counteract the increasing resistance of cancer cells to the available anticancer agents. Nature, especially the marine environment, has been and will continue to be the primary source of bioactive compounds against novel targets in many diseases including cancer. 

Cembranolides are cembrane-type diterpenoids that were organically isolated from marine organisms [9]. They are characterized by a fourteen-membered carbocyclic ring skeleton with a five-, six-, seven-, or eight-membered lactone ring. They attracted attention in the last few years due to their unique structural features and biological activities [10]. The cembrane derivatives of the soft coral *Lobophytum* species exhibited several pharmacological effects including antivirus, immunostimulatory, anti-inflammatory, anticancer, and antibacterial activities [11,12,13,14,15]. In our previous study, we found that the cembrane-type diterpenoids, 13-acetoxysarcocrassocolide, sarcocrassocolide M, and 14-deoxycrassin, exhibited potent cytotoxic activity against leukemia and lymphoma cells with IC_50_ values ranging from 1.2 to 7.1 μM [16]. A search in PubMed (August 2019) using the search term “13-acetoxysarcocrassolide” resulted in the retrieval of four peer-reviewed publications reporting the cytotoxic activity of this compound. By analyzing these publications, we found that 13-AC from the soft coral, *Sarcophyton crassocaule*, exhibited a wide range of cytotoxic properties against leukemia P388 cells [10], bladder female transitional cancer (BFTC) cells [17], and AGS (human gastric adenocarcinoma cells) gastric carcinoma cells [18]. It also showed an immunomodulatory effect on dendritic cells [15]. However, the precise cytotoxic mechanism of action of this natural product was not elucidated. This study aimed to evaluate the cytotoxic potential of 13-AC against oral cancer Ca9-22 cells along with its mechanism of action using in vitro cellular and in vivo xenograft models.

## 2. Results

### 2.1. Effect of 13-AC on Cellular Proliferation, Migration and DNA Damage in Oral Cancer Cells

Previous studies found that the extract of aquaculture *Lobophytum crissum* (Figure 1A) contained a variety of cembranoids, including 13-acetoxysarcocrassocolide (13-AC), deoxybassin and lobocrassin B [16]. Although previous studies showed that 13-AC (Figure 1B) exhibited potent cytotoxicity against bladder cancer, gastric cancer, leukemia, and lymphoma cell lines [16,17,18], it was not clear if this compound is effective in suppressing the proliferation of other cancer cell lines including oral, prostate, colon, breast, and cervical cancers. To reveal the antiproliferative activity of 13-AC, we tested its effect on several cancer cell lines using MTT cell proliferation assay. After 72 h of treatment, the IC_50_ values of 13-AC against Ca9-22, Cal-27, HCT116, Lovo, DLD-1, DU145, LNcap, MCF-7, T47D, and Hela cancer cells were 0.94 ± 0.16, 1.31 ± 0.38, 1.36 ± 0.27, 1.38 ± 0.37, 1.64 ± 0.36, 4.85 ± 0.92, 3.93 ± 1.06, 2.44 ± 0.22, 2.00 ± 0.09, and 4.41 ± 0.75 μg/mL, respectively. We next sought to evaluate the cytotoxic effect of cisplatin, a well-known chemotherapeutic drug, on the proliferation of these cancer cells. In contrast, cisplatin possessed a moderate inhibition of the cellular growth against Lovo, MCF-7, and T47D cells at the highest concentration (10 μg/mL). The data show that 13-AC was more potent than cisplatin against all tested cancer cell lines, except prostate cancer cells (Table 1). The oral Ca9-22 cancer cells were the most susceptible cancer cell lines to 13-AC, and thus they were selected for further in vitro and xenograft tumor growth in vivo studies.

To analyze the cytotoxicity of 13-AC against Ca9-22 oral cancer cells, the cells were treated with increasing concentrations (0.625, 1.5, 2.5, 5, and 10 µg/mL) of 13-AC for 24, 48, and 72 h and their proliferation was measured by MTT assay. The results confirm that the exposure to a lower concentration (1.25 µg/mL) of 13-AC resulted in a significant growth reduction in the treated cells up to 20%, 60%, and 80% after 24, 48, and 72 h, respectively, compared with the untreated cells. The treatment with 5 µg/mL of 13-AC produced up to 60–80% inhibition after 24, 48, and 72 h, and this inhibitory effect was increased to 90% inhibition with 10 µg/mL of 13-AC (Figure 1C). 13-AC exhibited potent cytotoxicity against Ca9-22 oral cancer cells in a concentration- and time-dependent manner.

To analyze the effect of 13-AC on the migration of oral cancer cells, Ca9-22 cells were treated with increasing concentrations (1.25, 2.5, and 5 µg/mL) of 13-AC for 6 and 12 h and the cellular migration was evaluated by wound healing assay. At concentrations of 1.25, 2.5, and 5 µg/mL, 13-AC reduced the wound closure by 37%, 48%, and 52%, respectively, after 12 h (Figure 2A). Based on this observation, 13-AC significantly suppressed Ca9-22 cells migration in a concentration- and time-dependent manners. It was confirmed that cancer cell migration is a characteristic feature of tumorigenesis. epithelial-to-mesenchymal transition (EMT) is a phenomenon that describes the change of an epithelial phenotype to a more motile and invasion mesenchymal phenotype and this phenomenon contributes to the pathogenesis of human carcinomas [19]. To assess the impact of 13-AC on EMT, Ca9-22 cells were treated with 10 µg/mL of 13-AC at the indicated time intervals. As shown in Figure 2B, the treatment of Ca9-22 cells with 13-AC resulted in the loss of the characteristic mesenchymal morphological features. We further studied whether the molecular mechanism of the 13-AC anti-migration effect involves the regulation of the HIF1/MMP/E-cadherin and snail pathway. In a time-dependent manner, 13-AC (10 µg/mL) treatment suppressed the expression of HIF-1, MMP 2, and slug, while the expression of E-cadherin was significantly increased as demonstrated by Western blotting analysis (Figure 2C). Next, we investigated whether 13-AC growth inhibitory and antimigration activities affect the DNA integrity of Ca9-22 cancer cells. We used a comet assay under neutral electrophoresis condition. In a concentration-dependent manner, 13-AC (2.5, 5, and 10 µg/mL) promoted DNA fragmentation in the cancer cells which was demonstrated by an increase of DNA migration (Figure 2D). The treatment with 13-AC (10 µg/mL) accelerated the phosphorylation of H2A.X at serine 139 (γ-H2A.X) after 9 and 12 h (Figure 2E). Our results indicate that the cytotoxic and antimigration effects of 13-AC were associated with the induction of nuclear DNA damage of Ca9-22 cells.

### 2.2. 13-AC Induces Apoptosis in Ca9-22 Cancer Cells

Previous studies showed that the marine cembranoid 13-AC induced apoptosis in the hepatic, bladder, and nasopharyngeal carcinomas [17,20,21]. Our results show that 13-AC exhibited antiproliferative and antimigration activities as well as induced DNA damage in Ca9-22 cells (Figure 1 and Figure 2). We then moved further to investigate if 13-AC can induce apoptosis in Ca9-22 cancer cells using flow cytometry and microscopy. The assessment of DAPI staining with a fluorescent microscope suggested that the nuclear morphological changes in Ca9-22 cells were induced with different concentrations of 13-AC after 24 h. The number of condensed nuclei in 13-AC-treated cells was much higher than the control that showed intact and normal nuclei (Figure 3A). The evaluation of annexin V-FITC/PI staining with flowcytometric analysis suggested that 13-AC significantly increased the apoptotic cell population (9% to 80%) in a concentration-dependent manner after 24 h (Figure 3B). To further examine the underlying apoptotic mechanism of 13-AC, the cleavage of caspases-3 and -9, as well as PARP, was examined with Western blotting. 13-AC promoted the cleavage of these proteins (Figure 3C). According to the previous study, specific caspase inhibitors (Z-LEHD-FMK and Z-DEVD-FMK) attenuated the apoptosis-induced 13-AC in AGS cells [18]. Whether specific pancaspase inhibitor, Z-VAD-FMK, affected viability and migration inhibition of 13-AC in Ca9-22 cells was investigated with MTT and wound healing assay. Unexpectedly, inhibition of viability and migration induced by 13-AC was not attenuated in the presence of Z-VAD-FMK (Figure 3D,E). Collectively, these results reveal that 13-AC induced caspase-independent apoptosis and antimigration in oral cancer Ca9-22 cells.

### 2.3. Effect of 13-AC on Mitochondrial Membrane Potential, ROS Generation, and Calcium Accumulation

Accumulating evidence shows that the disruption of calcium cellular concentration, mitochondrial membrane potential, and ROS homeostasis contribute to the induction of apoptosis [22,23,24]. Therefore, we examined whether 13-AC-induced apoptosis involves the disruption of calcium cellular concentration, mitochondrial membrane potential (MMP), and ROS homeostasis in Ca9-22 cells. We first assessed the disruption of mitochondrial membrane potential in Ca9-22 cells with JC-1 staining. The JC-1 staining showed accumulation in the mitochondria, and its fluorescence emission changed from green (~529 nm, FL1) to red (~590 nm, FL2). After 24 h, 13-AC resulted in a concentration-dependent (0, 2.5, 5, and 10 μg/mL) increase of the population of cells with disrupted mitochondrial membrane potential from 4% to 95% (Figure 4A). The generation of the intracellular ROS by 13-AC treatment was determined with a carboxyl derivative of fluorescein, carboxy-H_2_DCFDA dye using flow cytometric analysis [25]. As shown in Figure 4B, the treatment with 13-AC (10 μg/mL) for 0.5, 1, 3, and 6 h resulted in 3.32-, 3.96-, 4.98-, and 3.05-fold increases in ROS levels, respectively, as compared with the mean fluorescence index (MFI) of the control. ER stress can be induced by oxidative stress, leading to mitochondria-dependent apoptosis [26]. Our results indicate that the calcium accumulation was increased by 4.51, 4.43, 3.53, and 6.93 folds in 13-AC (10 μg/mL)-treated Ca9-22 cells after 1, 3, 6, and 18 h, respectively, as demonstrated by staining with a fluorescent calcium indicator, Fluo 3 (Figure 4C). Together, these results indicate that 13-AC disrupted mitochondrial integrity, the homeostasis of calcium, and ROS, resulting in apoptosis. Since oxidative stress has emerged as an attractive target for cancer therapy [27,28], we investigated whether apoptosis induction by 13-AC was due to the promotion of ROS generation. To examine this hypothesis, Ca9-22 cells were pretreated with 0.09, 0.375, and 1.5 mM of N-acetyl-L-cysteine (NAC), a ROS scavenger agent, aiming to neutralize the intracellular ROS level. Flow cytometric results reveal that the pretreatment of C9-22 cells with 0.09, 0.375, and 1.5 mM NAC inhibited 13-AC-induced apoptosis by 68.1%, 32.8%, and 4.0%, as well as the disruption of MMP by 45.8%, 24.9%, and 12.5%, at 10 µg/mL, respectively, (Figure 4D,E). The results of annexin V/PI and JC-1 staining suggest that the 13-AC-induced apoptosis and mitochondrial dysfunction resulting from ROS overproduction were suppressed by NAC pretreatment.

### 2.4. Oxidative Stress-Induced by 13-AC Involves the Dysregulation of Nrf2/Keap1/p62/SQSTM1 Pathway

The Keap1/Nrf2/p62/SQSTM1 signaling pathway is a major defense mechanism for oxidative stress [29,30,31]. Recent studies indicated that p62/SQSTM1 is a target gene of the transcription factor Nrf2 [32,33]. Under stressful conditions, phosphorylated p62 shows a higher affinity for Keap1, indicating that the phosphorylation of p62 causes disruption of Keap1-mediated Nrf2 ubiquitination [34]. To clarify if the underlying mechanism of 13-AC-induced Ca9-22 cellular oxidative stress involves the dysregulation of Nrf2/Keap1/p62/SQSTM1 signaling pathway, Western blotting, immunofluorescence, and immunoprecipitation were used to identify the target proteins defending Ca9-22 cells from ROS burden induced by 13-AC treatment. The expressions of Keap1, Nrf2, and p62 proteins were determined by Western blotting after the treatment with 10 μg/mL of 13-AC for the indicated time intervals. The treatment with 13-AC (10 μg/mL) for 3, 6, 9, and 12 h resulted in an increase of Nrf2 and p62 as well as a suppression of Keap 1 expression as compared with the control (Figure 5A). To evaluate the distribution of p62 protein, Ca9-22 cells were cultured with or without 13-AC (5 μg/mL) for 24 h and the effect was evaluated using confocal microscopy. As shown in Figure 5B, 13-AC (5 μg/mL) promoted the accumulation of p62 in the cytosol and nuclear fraction. In the p62–Keap1–Nrf2 axis, p62 acts as a modulator of Nrf2 activation. Growing evidence shows that the Keap1-interacting region (KIR) of p62 binds to Keap1 similar to the ETGE motif of Nrf2, thus preventing Keap1 from trapping Nrf2, which results in Nrf2 stabilization and activation [32,33,35,36]. We further investigated whether the direct interaction between p62/Nrf2 or p62/Keap1 disturbed the antioxidant functions of the p62/Keap1/Nrf2 axis after 13-AC treatment. The immunoprecipitation assay showed that p62 interacted with Nrf2 and Keap1 in a concentration-dependent manner after treatment with 5 and 10 μg/mL of 13-AC (Figure 5C). These results suggest that p62 interacted with Keap1/Nrf2 and formed complexes following 13-AC treatment. The expression of p62 was upregulated after 13-AC treatment suggesting that p62 may be an important target in the apoptosis induced by 13-AC. To verify whether p62 plays a crucial role in 13-AC induced apoptosis, MTT assay was performed to determine the cytotoxicity of 13-AC against Ca9-22 cells after the knockdown of p62. Our results show that the transfection of p62 siRNA protected Ca9-22 cells from the cytotoxic effect of 13-AC (5 and 10 μg/mL) by 10% to 49% and 7% to 39%, respectively (Figure 5D). The Western blotting assay was performed to further identify whether NAC pretreatment could attenuate the cleavage of PARP and activation of caspase-3 as well as the inactivation of the antioxidative-related proteins following 13-AC treatment (5 and 10 μg/mL). NAC pretreatment attenuated PARP cleavage, caspase-3 activation, caspase-9 activation, HO-1, Nrf2, and p62 expression but increased Keap1 expression despite the treatment with 5 and 10 μg/mL of 13-AC. These results were observed with Western blotting (Figure 5E). These results suggest that 13-AC induced apoptosis via ROS generation, which involved the dysregulation of the p62/Keap1/Nrf2 signaling pathway.

### 2.5. Antitumor Effect of 13-AC on Human Oral Ca9-22 Cancer Cells Xenograft Animal Model

An animal model was used to further evaluate the antitumor effect 13-AC against Ca9-22 cells following the potent cytotoxic results demonstrated by MTT and flow cytometric assays. Xenograft animal model with human oral Ca9-22 cancer cells was used to evaluate the in vivo antitumor activity of 13-AC. Ca9-22 cells (1 × 10^6^) were injected in the right flank of male immunodeficient athymic mice. The dose in the in vivo experiment of 13-AC was established based on the cytotoxicity results of 13-AC leading to 50% inhibition of cell proliferation after 72 h. Based on the in vitro study, we used 4 mg/kg. 13-AC significantly inhibited tumor growth. The mice were treated for 30 days and the average tumor volume in the control group was 519 ± 171 mm^3^, whereas the average tumor volume in the 13-AC-treated group was 241 ± 141 mm^3^ (Figure 6A). As compared with the control group, the tumor volume was significantly suppressed in the 13-AC groups by 53.56%. The mice’s body weights did not show significant changes and no histopathological changes of the heart, liver, kidney, lung, and spleen, which were recognized by H&E staining (Figure 6B,F). At the end of the treatment, the tumor tissues were isolated and weighed demonstrating smaller average weights in the 13-AC-treated groups (20 mg) compared with the control group (250 mg) (Figure 6C,D). Tumor cells showed spindle squamous epithelial cells, high nucleic/cytoplasm ratio with high mitosis, and keratin pears (arrow) in the solvent control. In the 13-AC groups, the tumor cells were smaller, atrophic, and showed less survival (Figure 6E). In the central area, extensive necrosis and cystic formation were also detected. The tumor volume was reduced with no significant changes in the mice weights. The mice tissues showed no significant histological differences. These findings suggest that, in the in vivo xenograft model, 13-AC displayed a potent anti-tumor activity without significant side effects.

## 3. Discussion

13-Acetoxysarcocrassolide (13-AC) was first isolated from the Formosan soft coral *Sarcophyton crassocaule* and was identified as a cembrenolide diterpene [10]. 13-AC exhibited a cytotoxic effect on bladder and stomach cancer cell lines by inducing apoptosis through the regulation of hnRNPs F/H and PPT1 as well as the activation p38/JNK pathway and the inhibition of PI3K/Akt growth pathways [17]. Our previous studies showed that 13-AC is an interesting compound from different perspectives including its unique chemical structure with specific functionality of α-methylene-γ-lactone, the potent cytotoxic effect against several cancer cell lines, and the high yield isolated from the cultured coral, *Lobophytum crassum* [16]. These compelling features encouraged us to work on this unique cytotoxic marine compound. We initially examined the antiproliferative effect of 13-AC on the growth of several cancer cell lines including leukemia, colorectal cancer, oral cancer, adenocarcinoma, breast cancer, and cervical cancer by MTT assay. It showed potent cytotoxicity against most of the cancer cell lines compared with cisplatin (Table 1). We then examined its antitumor effect by inoculating human oral cancer Ca9-22 cells in xenograft animals. The results of the in vivo experiment also confirmed that 13-AC not only reduced the size and weight of the tumor but also did not affect the body weight of the mice (Figure 6A–F). The values of the blood biochemical tests were within the permissible range and suggested no adverse effects on the liver and kidney functions (data are not shown).

Head and neck squamous cell carcinoma (HNSCC) are among the deadliest cancers resulting in over 300,000 deaths worldwide. Most HNSCC patients seek medical help in the late stage of the disease (stage III/IV) and usually have a five-year survival rate [37]. The activation of NF-κB, change of epithelial-to-mesenchymal transition (EMT), and cell adhesion deregulation were identified as prominent genetic alterations in HNSCC development and/or progression [38]. The process of EMT can alter the polar and adhesive phenotype of the epithelial tumor cells to a mesenchymal phenotype, resulting in promoting cell migration, invasion potential, cytoskeleton remodeling, and resistance to apoptosis [39,40]. The low expression of E-cadherin and high expression of vimentin are correlated with an increase of distant metastasis in primary HNSCC tumors [41]. In wound healing assay and morphologic observation, the 13-AC treatment showed a significant suppression in cell migration and inhibition of EMT morphological changes in C9-22 cells (Figure 2A,B). Our findings show that the treatment of 13-AC increased the expression of E-cadherin and decreased the expression of HIF-1, MMP-2, and slug in C9-22 cells after 9–12 h of treatment (Figure 2C). On the other hand, 13-AC promoted the expression of vimentin (data not shown).

Modulation of oxidative stress is an important component of anticancer therapy. Currently, several FDA-approved anticancer drugs with direct effects on ROS are used for effective cancer treatment [42]. NOV-002, a glutathione disulfide mimetic, alters intracellular GSSG/GSH ratio in lung and breast cancers [43]. Sulfasalazine, an inhibitor of cystine/glutamate transporter XCT, reduces intracellular transport of cysteine required for GSH synthesis in pancreatic and lung cancers [44]. Celecoxib, an inhibitor of cyclooxygenase 2 (COX2), and nelfinavir, an HIV protease inhibitor, both induce ER stress by causing leakage of calcium from the ER into the cytosol via the induction of ROS in colorectal cancer, myeloma, and prostate cancer as well as HPV-transformed cervical carcinoma, head and neck cancer, pancreatic cancer, and melanoma [45,46]. *N*-Benzyloxycarbonyl-Ile-Glu (*O*-t-butyl)-Ala-leucinal (PSI), a proteasome inhibitor, induces ROS resulting in mitochondrial dysfunction in neuronal cells [47]. 5-Fluorouracil (5-FU), an inhibitor of thymidylate synthetase, induces an intracellular increase in O_2_^-^ levels in colon, rectum, and head and neck cancers [48]. Gambogic acid was found to sensitize ovarian cancer cell line (SKOV-3) to doxorubicin via ROS-mediated apoptosis [49]. Understanding the molecular mechanism involved in the induction of oxidative stress in cancer cells can lead to the development of novel anticancer drugs.

Chemical stimulation of cancer cells may cause apoptosis through the destruction of mitochondrial function, induction of ER stress, and the overproduction of reactive oxygen species resulting in the irreversible death of the cells [50]. In this study, we investigated the therapeutic potential of the marine cembrenolide, 13-AC, on human oral cancer C9-22 cells. Flow cytometric analysis showed that the antitumor effect of 13-AC induced ROS generation, ER stress, mitochondrial dysfunction, and apoptotic cell death of Ca9-22 cells (Figure 3B and Figure 4A–E). Previous studies suggested that the antioxidative function of Nrf2 prevented carcinogenesis by quenching ROS or repairing oxidative damage [51,52], while other reports suggested that Nrf2 activation promotes detoxification and tumorigenesis in cancer cells [52,53,54]. We observed that 13-AC treatment initially stimulated reactive oxygen species (ROS) generation and perturbed the p62/Keap1/Nrf2 signaling pathway. The results of Western blot show that 13-AC treatment attenuated the expression of Keap1 as well as elevated the expression of p62 and Nrf2 (Figure 5A). Moreover, the immunofluorescent assay showed that 13-AC treatment promoted the expression of p62 leading to its accumulation in the nuclear compartment of Ca9-22 cells (Figure 5B). Previous reports showed that p62 docks directly onto the Keap1, via a motif, designated the Keap1 interacting region (KIR), thereby blocking the binding between KEAP1 and NRF2 that leads to ubiquitylation and the degradation of the transcription factor [55]. However, p62 enhanced the interaction with Nrf2 but decreased the interaction with Keap1 in Ca9-22 cells treated with 13-AC (5 and 10 µg/mL), as demonstrated by the immunoprecipitation analysis (Figure 5C). Nrf2 is a cytoprotective gene and can drive the expression of antioxidant enzymes, heme oxygenase (HMOX1), NAD(P)H: quinone oxidoreductase 1 (NQO1), glutathione S-transferases (GSTs) and UDP-glucuronosyltransferases (UGTs) to scavenge ROS under oxidative stress [56]. In this study, 13-AC treatment promoted Nrf2 and p62 protein expression and induced ROS generation. This could be explained by p62 interaction with Keap1 and Nrf2 proteins resulting in the inactivation of gene transcription of Nrf2 and causing irreversible oxidative stress leading to cell apoptosis. 

A previous study proposed that p62 upregulated the interaction and ubiquitination of Nrf2 along with the reduction of Keap1 expression, thereby increased the expression of p62, which was consistent with the enhancement of apoptosis [57]. According to previous studies, the dysregulation of the p62/Keap1/Nrf2 axis affected cancer development [58]. Nrf2 is considered as a tumor suppressor. Nrf2 inhibitors can provide a clear rationale for the consideration of Nrf2 as a powerful putative therapeutic target in cancer treatment [59]. Despite being discovered a long time ago, the functional properties of the p62/SQSTM1/Keap1/Nrf2 pathway remained poorly understood. It is, therefore, crucial to gain deeper insights into the function of the p62/SQSTM1/Keap1/Nrf2 pathway and the underlying mechanisms. In this axis, p62 acts as a modulator of Nrf2 activation [35]. However, Nrf2 is highly expressed in lung, head and neck, breast, and ovarian cancer cells. It promotes metastasizes of cancer cells, angiogenesis, and even resistance to chemotherapy [31,60]. p62 is a stress-inducible protein, but it is also a signal protein linked to the transcription factor NFκB. The overexpression of p62 may inhibit or promote the activation of NFκB. It was demonstrated that the activation of p62 can affect the external pathway of apoptosis, whereas p62 can regulate cell autophagy in tumorigenesis rendering it a key factor in apoptosis and autophagy [61,62]. According to previous studies, p62 can cause apoptosis by activating caspase 8. Increasing evidence demonstrates that p62 acts as a signal hub for conduction mammalian target of rapamycin complex 1 (mTORC1) and affects the Keap1/Nrf2 pathway to induce apoptosis or autophagy [63,64]. Our data demonstrate that the knockdown of p62 attenuated cytotoxicity of 13-AC in Ca9-22 cells (Figure 5D), validating the important role of p62 in 13-AC-induced apoptosis. Our findings suggest that the p62/SQSTM1/Nrf2 pathway can be a potential target against oxidative stress-induced by chemotherapeutics. 

In summary, our study revealed that 13-AC isolated from the coral aquaculture of *Lobophytum crassum* exhibited a potent antitumor effect by inducing mitochondrial-dependent apoptosis in human oral cancer Ca9-22 cells. The pretreatment with ROS scavenger, NAC, not only attenuated the apoptotic effect of 13-AC but also abrogated the effect of 13-AC on the disruption of mitochondrial membrane potential. We confirmed the results published earlier in a study on gastric carcinoma cells reporting that 13-AC could induce mitochondria-related apoptosis via p38/JNK activation and PI3K/AKT suppression [18]. Such notable induction of mitochondrial apoptosis will predict that increasing priming of cancer cells would make them more sensitive, potentially offering a new therapeutic window. Collectively, these results demonstrate that the marine cembranolide, 13-AC, stimulated oxidative stress that promoted the dysregulation of the p62-Keap1-Nrf2 signaling pathway. This is the first study to report the potential development of 13-AC as a potent Nrf2 inhibitor in cancer therapy, especially against oral cancer.

## 4. Materials and Methods 

### 4.1. Cell Lines, Biological Materials and Chemicals 

The American Type Culture Collection was the source of the used cell lines (ATCC, Manassas, VA, USA). The cell lines were kept in 5% CO_2_ humidified atmosphere at 37 °C. RPMI 1640 was the growing medium for DLD-1, T47D, and LNcap cells. Dulbecco’s Modified Eagle’s was the growing medium for Ca9-22 and Cal-27 cells. Minimum essential medium was the growing medium for DU145, MCF-7, and Hela cells. McCoy’s 5a was the growing medium for HCT116 cells. Ham’s F-12 medium was the growing medium for Lovo cells. Glutamine (2 mM), antibiotics (100 μg/mL of streptomycin and 100 units/mL of penicillin), and fetal calf (10%) serum were used to supplement the growing media. Cell Signaling Technologies (Beverly, MA, USA) was the source of antibodies against caspases-3 (#9662) and -9 (#9052), cleaved PARP (#5625), p62 (#8025), MMP-2 (#4022), Nrf2 (#12721), and keap1 (#8047). GibcoBRL (Gaithersburg, MD, USA) was the supplier for trypan blue, streptomycin, RPMI 1640 medium, fetal calf serum (FCS), and penicillin G. Cisplatin (ALX-400-040-M250) and HIF1 (Halpha111a) antibody were purchased from Enzo life sciences (Taipei, Taiwan). Santa Cruz Biotechnology (Santa Cruz, CA, USA) was the source of antibodies for p-H2AX, E-cadherin (sc-31021), HO-1 (sc-136960), slug (sc-166476), and Nrf2 (sc-365949). Pierce (Rockford, IL, USA) was the supplier for anti-mouse and rabbit IgG peroxidase-conjugated secondary antibody. Strong Biotech Corporation (Taipei, Taiwan) was the supplier of annexin V-FITC/PI (propidium iodide) stain. Sigma-Aldrich (St. Louis, MO, USA) was the supplier of dimethyl sulfoxide (DMSO), 3-(4,5-dimethylthiazol-2-yl)-2,5-diphenyl-tetrazolium bromide (MTT), and all other chemicals. Molecular Probes and Invitrogen technologies (Carlsbad, CA, USA) was the source of JC-1 cationic dye, the carboxy derivative of fluorescein (carboxy-H_2_DCFDA), and Fluo-3. ECL. Amersham Life Sciences (Amersham, UK) was the supplier for the Hybond ECL transfer membrane and Western blotting detection kits.

### 4.2. The Stock Solution of 13-AC

13-AC was separated from the coral *Lobophytum crissum*. The spectroscopic data (1D and 2D NMR) of the isolated compound were compared to those reported in the literature, and the chemical structure of 13-AC was confirmed [16]. A stock solution of 13-AC was prepared by dissolving the compound in DMSO (dimethyl sulfoxide) (20 µg/mL) and a series of dilutions was prepared before use.

### 4.3. MTT Cell Proliferation Assay

In the MTT assay, culture plates with 96-well were used. Seeding was done at 4 × 10^4^ per well and the tested materials were added to the cells at different concentrations [65,66]. The MTT assay (thiazolyl blue tetrazolium bromide, Sigma-M2128) was used to examine the cytotoxic effect of the extract after 24, 48, or 72 h. The light absorbance values at 570 and 620 nm were measured using ELISA reader (Anthoslabtec Instrument, Salzburg, Austria) (OD = OD_570_–OD_620_). The IC_50_ value (concentration that caused 50% inhibition) was calculated. The findings were articulated as the percentage of the control ± SD obtained from *n* = 4 wells per experiment from three independent experiments.

### 4.4. Annexin V/PI Apoptotic Assay

The membrane integrity and phosphatidylserine (PS) externalization were determined using annexin V-FITC staining kit [67]. Plates with 35-mm diameter were used to grow cells (10^6^). To label the cells, annexin V-FITC (10 μg/mL), and PI (20 μg/mL) were used. To wash all plates, a binding buffer was used and then the cells were harvested. To suspend the cells, a binding buffer was used (2 × 10^5^ cells/mL) and FACS-Calibur flow cytometer (Beckman Coulter, Taipei, Taiwan) was used to assess the label. CellQuest software was used to analyze the results. For each measurement, approximately 10,000 cells were counted.

### 4.5. Determination of ROS Generation, MMP Disruption, and Calcium Accumulation 

The determination of calcium accumulation, MMP disruption, and ROS generation was reported previously [24,65]. Fluorescent calcium indicator (Fluo 3, 5 mM), JC-1 cationic dye (1.25 μg/mL), and the carboxy derivative of fluorescein (carboxy-H2DCFDA, 1.0 mM) were used to determine calcium accumulation, MMP disruption, and ROS generation, respectively. A specific fluorescent dye was used to label cells were labeled for 30 min after treating the cells with different concentrations of 13-AC. PBS was used for washing. Flow cytometry was used to determine the changes of ROS, MMP, and calcium concentrations. 

### 4.6. Protein Immunoprecipitation and Western Blot Analysis

RIPA lysis buffer (1% Nonidet P-40, 1× PBS, 0.1% sodium dodecyl sulfate (SDS), 0.5% sodium deoxycholate, 1 mM sodium orthovanadate, 100 μg/mL phenylmethylsulfonyl xuoride, and 30 μg/mL aprotinin) was added to the cells for 30 min to obtain cell lysates [66]. A centrifugation was performed for the lysates at 20,000× *g* for 30 min. BCA protein assay kit (Pierce, Rockford, IL, USA) was used to determine the protein concentration in the supernatant. Equal amounts of proteins were separated using SDS–polyacrylamide gel electrophoresis (7.5%, 10%, or 12%), which were then electrotransferred to a PVDF membrane. To block the membrane for 1 h, a solution containing 5% nonfat dried milk TBST buffer (20 mM Tris–HCl, pH 7.4, 150 mM NaCl, and 0.1% Tween 20) was used and the membrane was then washed with TBST buffer. The protein expressions were monitored using an immunoblotting assay with specific antibodies. To detect these proteins, an enhanced chemiluminescence kit (Pierce, Rockford, IL, USA) was used.

### 4.7. Immunofluorescence Analysis

The cells were treated with the tested compound. Paraformaldehyde (4%) in 50 mM HEPES buffer (pH 7.3) was used to fix the cells for 30 min. Triton X-100 (0.2%) in PBS (pH 7.4) was used to permeabilize the cells for 20 min. BSA (5%) in PBS containing 0.05% Triton X-100 (T-PBS) was the incubation solution for the cells for 1 h. The experiment was performed at room temperature to avoid non-specific protein binding. The primary antibodies (1:1000) was added to the cells and were incubated for 2 h. Dilutions were made for the secondary antibodies (Alexa Fluor 586-conjugated goat anti-mouse IgG or Alexa Fluor 488-conjugated goat anti-rabbit IgG) (Life Technologies, Carlsbad, CA, USA) at 1:1000 for 1 h at room temperature. The cells were washed with PBS. FV1000 confocal laser scanning microscope (Olympus, Tokyo, Japan) was used to examine the cells. 

### 4.8. The Detection of DNA Double-Strand Breaks (DSBs) Using Neutral Comet Assay For 

The determination of DNA double-strand-breaks was described in previous literature [50]. The assay was performed following the manufacturer’s protocol for the neutral comet assay utilizing a CometAssay^TM^ Kit (Trevigen, Gaithersburg, MD, USA). 13-AC at different concentrations (0, 2.5, 5, and 10 μg/mL) were added to the cancer cells (2 × 10^5^ cells/mL) for 3 h. Low melting point agarose (1%) was mixed with the cells were combined at a ratio of 1:10 (v/v). A portion of the mixture (75 μL) was added onto CometSlide^TM^ and left to settle at 4 °C in the dark. For 30–60 min, the slides were immersed in ice-cold lysis solution (Trevigen). A horizontal electrophoresis was used, and the slides were electrophoresed in 1× TBE (90 mM Tris-HCl, 90 mM boric acid, and 2 mM EDTA, pH 8.0) at 20 V for 10 min. To visualize cellular DNA, the samples were fixed in ethanol (70%) and dried before staining with 1:10,000 SYBR Green I (Trevigen). TriTek Comet Image program was utilized to analyze the fluorescence images to circumscribe the “head” and the “tail” regions of each comet. The integrated fluorescence values of each defined area were recorded. From the trailing edge of the nucleus to the leading edge of the tail, the comet length was determined. The comet length was used as an indication of the extent of the DNA damage. Calculations were averaged per replicate.

### 4.9. RNA Interference Transfection

Cancer cells were plated in a six-well plate and the manufacturer’s protocols in the serum-free Opti-MEM medium was followed to transfect the cells with 40 nM of ON-TARGET plus human p62 siRNA (GE Healthcare Dharmacon) using Lipofectamine^TM^ 2000 transfection reagent (Invitrogen, Carlsbad, CA, USA). The fresh medium with 10% FBS was replaced after 6 h. After 24 h of transfection, the experiments were performed. The study was approved by The National Museum of Marine Biology & Aquarium (NMMBA).

Sequences of p62/SQSTM1-ON-TARGET plus SMARTpool siRNA:

GAACAGAUGGAGUCGGAUA;GCAUUGAAGUUGAUAUCGA; CCACAGGGCUGAAGGAAGC; GGACCCAUCUGUCUUCAAA

### 4.10. Animal Material and Xenograft Animal Model with Human Oral Cancer Cells 

In nude mice, the xenograft animal model was established following the reported literature [24,25]. The National Laboratory Animal and Research Center (Taipei, Taiwan) was the supplier for the six-week-old male immunodeficient athymic mice. The mice were maintained under standard laboratory conditions (24–26 °C and 12–12 h dark–light circle). The mice were fed with a laboratory diet and water. The specimens of the aquaculture *Lobophytum crassum* soft coral were collected in 2015 as reported in our previous study [16]. At the National Museum of Marine Biology & Aquarium (Pingtung, Taiwan), the coral was preserved and aquacultured. The study was approved by the Animal Care and Treatment Committee of the National Museum of Marine Biology & Aquarium (NMMBA) (IACUC Permit Number 201706). Dealing with animals, the recommendations in the guidelines of the Care and Use of Laboratory Animals of the National Institutes of Health were followed. All efforts were executed to minimize animal stress/distress. Cancer cells (1 × 10^6^) were suspended in 0.2 mL PBS and then injected into the right flank of each mouse subcutaneously. The tumor growth was monitored every day. Mice with confirmed tumor growth, after fourteen days, were randomly divided into three groups. 13-AC (4 mg/kg) was injected intraperitoneally in the treatment group and the control group was administered only the solvent through intraperitoneal injection. For 30 days, the tested compound was administrated every other day. Carbon dioxide was used to kill the mice. A caliper was used to measure the tumor volume three times a week. The following equation width^2^ × length/2 was implemented to estimate the tumor volume.

### 4.11. Statistics

The findings are stated as the mean ± standard deviation (SD). An unpaired Student’s *t*-test was implemented to compare each experiment. A *p*-value of less than 0.05 was considered statistically significant.

## 5. Conclusions

Our previous study suggested that the two cembranoids 13-acetoxysarcocrassocolide (13-AC) and 14-deoxycrassin, from *L*. *crassum,* were responsible for its anti-proliferative activity. The presence of α-methylene-γ-lactone or α-methylene-*δ*-lactone moieties in these two cembranoids promoted their activity against leukemia cancer cell lines. In the present study, we demonstrated that 13-AC exhibited significant cytotoxic against several cancer cells with IC_50_ values of less than 4 µg/mL. Our results indicate that the treatment of cancer cells with 13-AC induced mitochondrial dysfunction and oxidative and ER stresses leading to apoptosis. Consequently, these cembranoid diterpenes possess unique structural features allowing them to preferentially inhibit the antioxidant function of Nrf2 and demonstrate their potential to be developed as anticancer drug leads.

## Figures and Tables

**Figure 1 marinedrugs-18-00382-f001:**
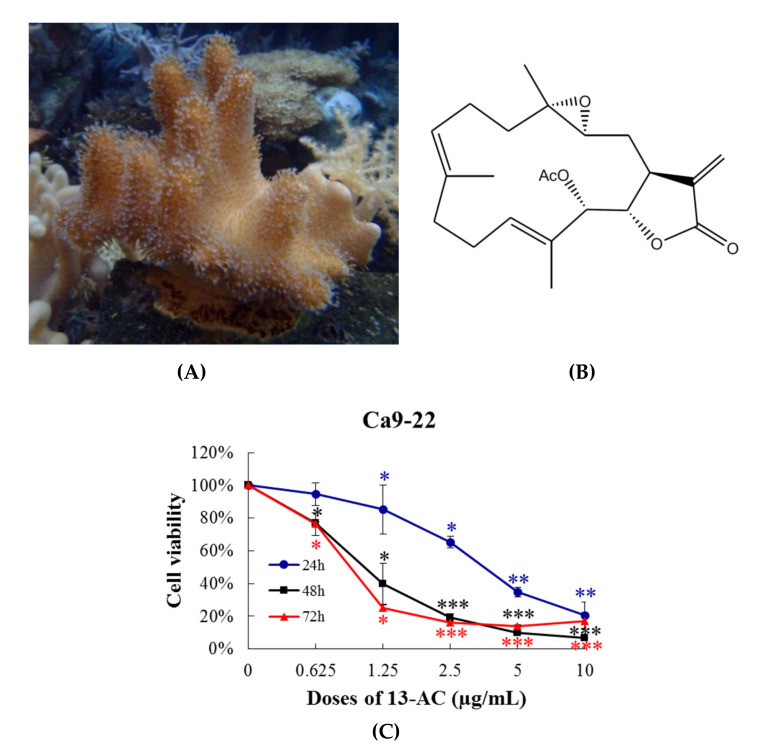
Effect of 13-AC on cellular viability in vitro MTT assay. (**A**) morphology of the aquaculture coral *Lobophytum crissum*; (**B**) chemical structure of 13-AC isolated from the coral *Lobophytum crissum*; and (**C**) human oral cancer cells Ca9-22 were treated with 13-AC at several concentrations for 24, 48, and 72 h and the antiproliferative activity was analyzed with MTT assay. The results are presented as the mean ± SD of three independent experiments. *, ** and *** (red, blue, black) were represented the statistic differences were denoted at * *p* < 0.05; ** *p* < 0.01; *** *p* < 0.001 when compared with the negative control.

**Figure 2 marinedrugs-18-00382-f002:**
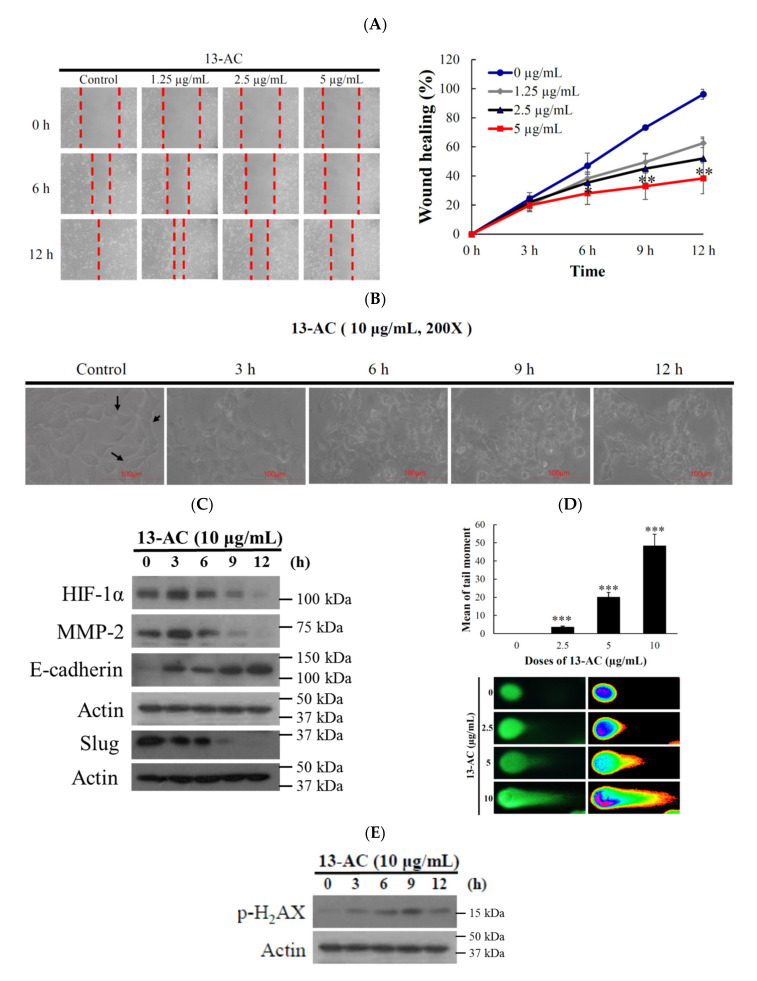
Effect of 13-AC on the migration and DNA damage of human oral cancer Ca9-22 cells. (**A**) Wound healing assay of Ca9-22 cells incubated with different concentrations of 13-AC. The results were evaluated by inverted optical microscopy (200×). Quantitative analysis of the relative cell migration was performed after 6 and 12 h. (**B**) Morphologic changes of epithelial-to-mesenchymal transition in Ca9-22 cells-incubated with 10 μg/mL of 13-AC was evaluated by inverted optical microscopy (200×). The changes of the relative cell migration were performed after 3, 6, 9, and 12 h. (**C**) Expression of migration-related proteins. (**D**) An example of “comet tail” due to chromosomal DNA double-strand breaks in 13-AC (2.5, 5 and 10 μg/mL)-treated Ca9-22 cells compared with the untreated control. Electrophoresis was carried out under neutral conditions. Quantitative results showed a gradual increase in tail movement with 13-AC treatment for 3 h in comparison with the control (200×). The results are presented as the mean ± SD of three independent experiments. (* *p* < 0.05; ** *p* < 0.01; *** *p* < 0.001, control vs. 13-AC group). (**E**) Biomarkers of DNA damage were determined with Western blotting assay. The cells were treated with 13-AC (10 µg/mL) for the indicated time intervals and were subjected to SDS-PAGE followed by Western blotting analysis. Actin was used as the loading control.

**Figure 3 marinedrugs-18-00382-f003:**
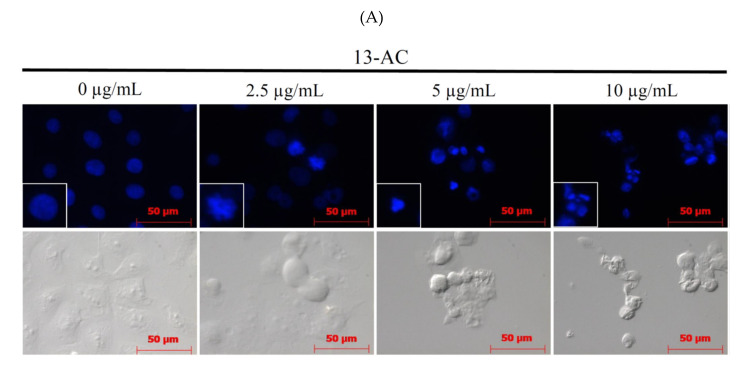
Effect of 13-AC on apoptosis induction in Ca9-22 cells. (**A**) The changes of nuclear morphology were determined with DAPI staining and were observed using a fluorescent microscope. (**B**) Apoptosis induction was assessed with annexin V/PI staining using flow cytometric analysis. (**C**) The effect of 13-AC on the expression of apoptosis-related proteins was determined with the Western blotting analysis. Actin was used as the loading control. To investigate whether the effect of the pretreatment with pan-caspase inhibitor (Z-VAD-FMK) on viability and migration was inhibited with 13-AC in Ca9-22 cells, (**D**) MTT and (**E**) wound healing assay were performed. The results are presented as means ± SD of three independent experiments. The statistic differences were denoted at * *p* < 0.05; ** *p* < 0.01; *** *p* < 0.001 when compared with the negative control.

**Figure 4 marinedrugs-18-00382-f004:**
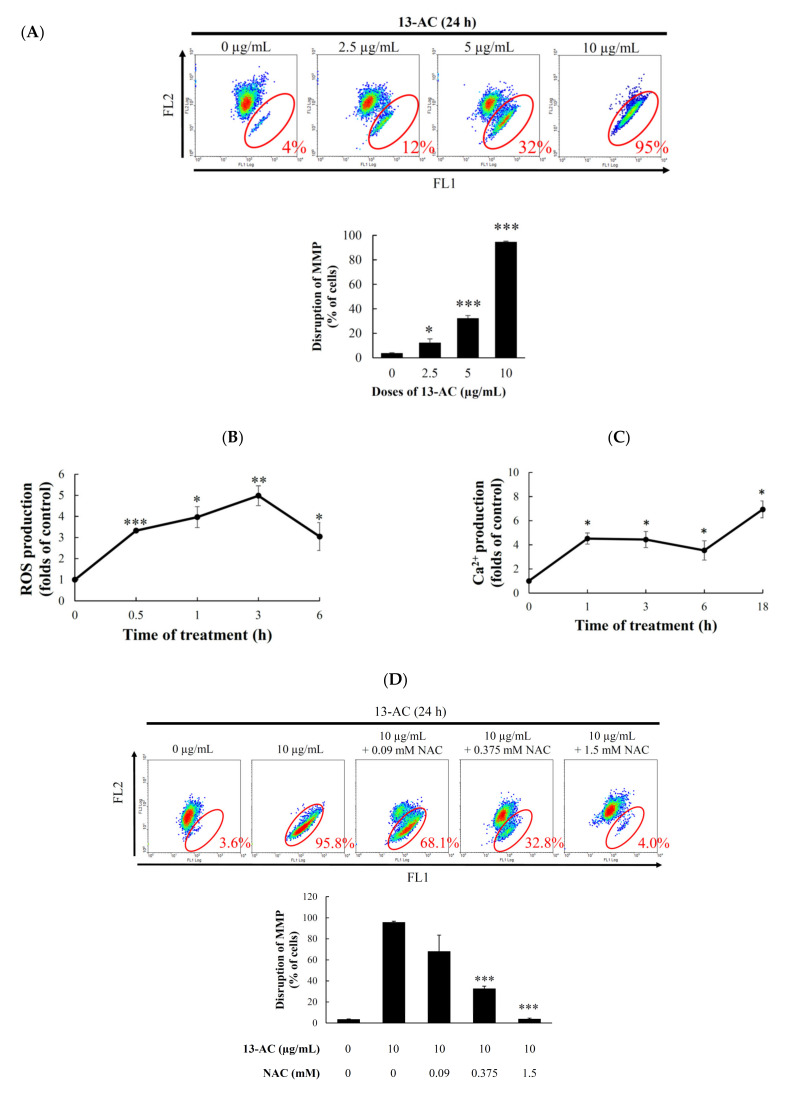
The apoptotic effect of 13-AC involves the induction of MMP disruption, ROS generation, and ER stress. (**A**) Cells were treated with the indicated concentration of 13-AC for 24 h. The disruption of mitochondrial membrane potential was evaluated by JC-1 staining using flow cytometric analysis. The results are presented as means ± SD of three independent experiments (* *p* < 0.05; *** *p* < 0.001), (**B**,**C**) The effect of 13-AC on ROS generation and calcium accumulation. Cells were treated with 13-AC (10 μg/mL) for the indicated times, respectively. Quantitative results show a gradual increase in ROS production or calcium accumulation in response to the 13-AC treatment when compared with the control group. (**D**) The disruption of MMP and (**E**) the apoptotic population was examined with JC-1 and annexin-V/PI staining, respectively. The apoptotic induction of 13-AC in Ca9-22 cells involves ROS production. Cells were pretreated with 0.09, 0.375, and 1.5 mM of NAC for 2 h, and then treated with 10 μg/mL of 13-AC for further 24 h. The results are presented as means ± SD of three independent experiments (* *p* < 0.05; ** *p* < 0.01; *** *p* < 0.001).

**Figure 5 marinedrugs-18-00382-f005:**
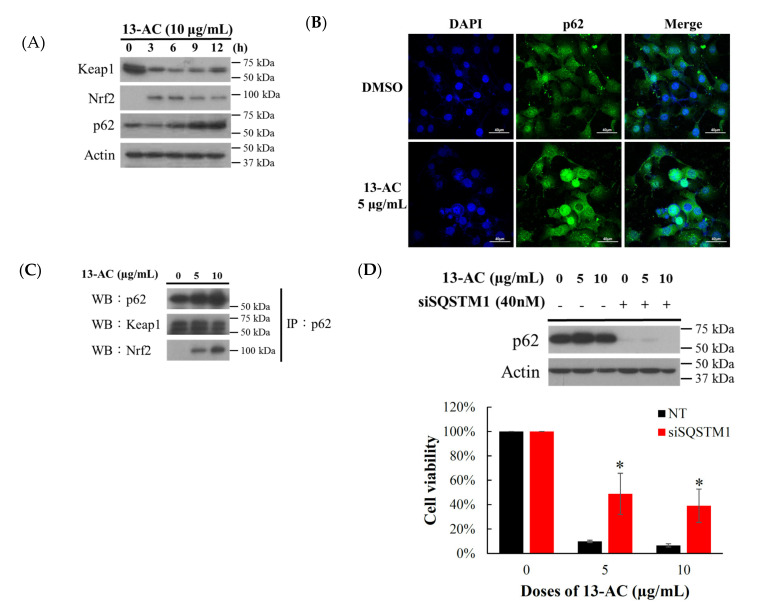
Effect of 13-AC on oxidative stress response pathway. (**A**) Effect of 13-AC on the expression of Keap-Nrf2 pathway-related proteins. (**B**) Effect of 13-AC on the localization of p62 protein examined by immunofluorescence confocal microscope. (**C**) After 13-AC treatment, the proteins were collected and immunoprecipitated with an anti-p62 specific antibody, and the samples were analyzed by Western blotting using specific antibodies for p62, Keap1, and Nrf2, respectively. (**D**) Effect of p62 on 13-AC-induced Ca9-22 cells apoptosis. Ca9-22 cells were transfected with siRNA targeting SQSTM1 (p62) for 24 h and then incubated with 5 and 10 μg/mL of 13-AC for further 24 h. Cell growth was evaluated with the MTT assay. The results are presented as means ± SD of three independent experiments (* *p* < 0.05). (**E**) Effect of NAC pretreatment on the cytotoxic activity of 13-AC was examined with Western blotting assay. Actin was used as the loading control.

**Figure 6 marinedrugs-18-00382-f006:**
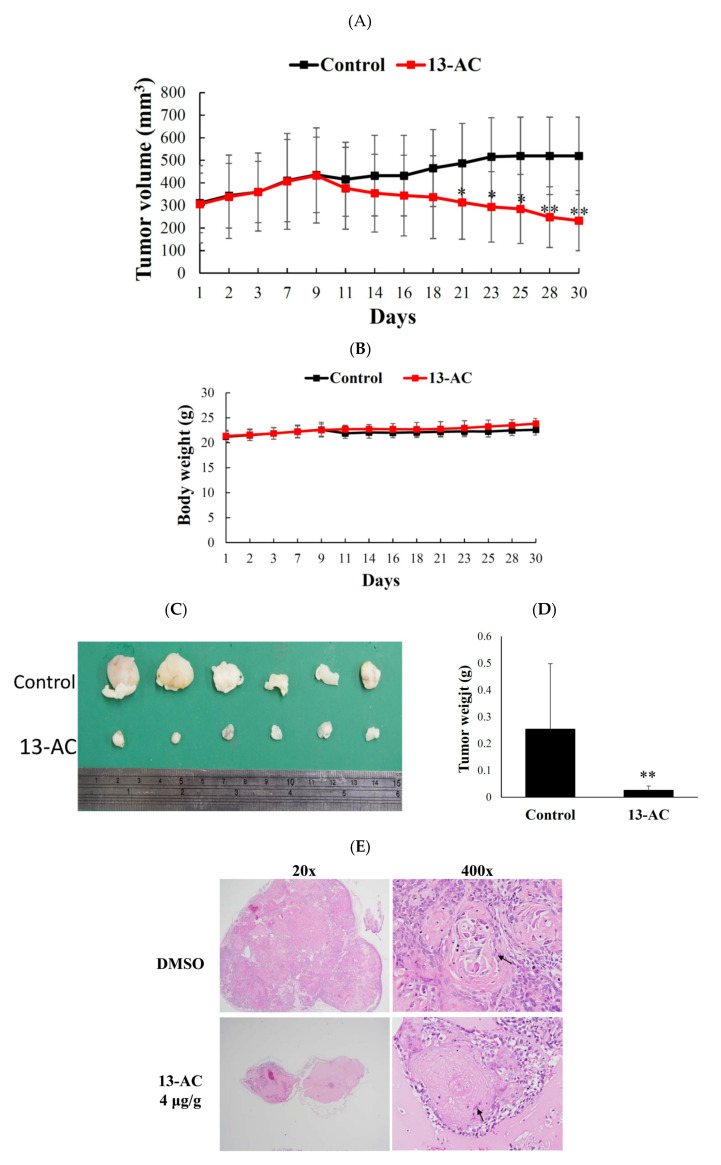
Effect of 13-AC on tumor growth and body weight in vivo human Ca9-22 tumor xenograft animal model. The solvent control (DMSO) or 13-AC (4 μg/g) was intraperitoneally injected in tumor-bearing nude mice for 30 days. (**A**) Every other day, the tumor volumes were measured, and the results are expressed as mean ± SD. * Significantly different from the control groups at * *p* < 0.05; ** *p* < 0.01. (**B**) The body weight was measured every other day, and the results are expressed as mean ± SD. Control, *n* = 10; 13-AC, *n* = 9. (**C**) The subcutaneous tumors representative photos that were collected after treatment with the solvent only (upper) or with 13-AC (lower) for 30 days. (**D**) Histogram of the tumor weight from the control group and 13-AC treated group. Values are expressed as mean ± SD. ** Significantly different from the control groups at *p* = 0.006. (**E**) Histopathological findings of DMSO and 13-AC in the subcutaneous xenograft of squamous Ca9-22 cells in mice using H&E stain (20× and 400×). The black arrow showed that the tumor cells expressed spindle squamous epithelial cells, high nucleic/cytoplasm ratio with highly mitosis, and keratin pears (arrow). (**F**) Histopathological findings of organs after the treatment with DMSO and 13-AC in the subcutaneous xenograft of squamous Ca9-22 cells in mice. No significant lesions in the heart, kidneys, liver, lungs, and spleen were recognized in the solvent control and 13-AC group stained with H&E stain (400×).

**Table 1 marinedrugs-18-00382-t001:** The effect of 13-AC on the viability of many cancer cell lines. The values of IC_50_ were calculated by calcuSyn software (Biosoft, Ferguson, MO, USA). The results are presented as the mean ± SD of three independent experiments.

Cancer Types	Cell Lines	13-AC IC_50_ (µg/mL)	Cisplatin IC_50_ (µg/mL)
Human oral cancer cells	Ca9-22	0.94 ± 0.16	3.47 ± 0.33
	Cal-27	1.31 ± 0.38	1.78 ± 0.17
Human colon cancer cells	HCT116	1.36 ± 0.27	3.62 ± 0.19
	LoVo	1.38 ± 0.37	NA
	DLD1	1.64 ± 0.36	8.79 ± 0.52
Human prostate cancer cells	DU145	4.85 ± 0.92	3.24 ± 0.36
	LNcap	3.93 ± 1.06	2.10 ± 0.20
Human breast cancer cells	MCF-7	2.44 ± 0.22	NA
	T47D	2.00 ± 0.09	NA
Human cervical cancer cells	HeLa	4.41 ± 0.75	7.74 ± 0.30

NA, not active at 10 µg/mL.

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
