# Peer review of "13-Acetoxysarcocrassolide Exhibits Cytotoxic Activity against Oral Cancer Cells through the Interruption of the Keap1/Nrf2/p62/SQSTM1 Pathway: The Need to Move Beyond Classical Concepts"

_marinedrugs, 2020, doi:10.3390/md18080382_

Round 1

Reviewer 1 Report

The manuscript is recommended for publication in the Marine Drugs.

Author Response

Reply to the Reviewers’ Comments

Manuscript ID: marinedrugs-850909

Manuscript Title: 13-Acetoxysarcocrassolide Exhibits Cytotoxic Activity Against Oral Cancer Cells Through the Interruption of the Keap1/Nrf2/p62/SQSTM1 Pathway: The Need to Move beyond Classical Concepts

Authors: Yi-Chang Liu, Bo-Rong Peng, Kai-Cheng Hsu, Mohamed El-Shazly, Shou-Ping Shih, Tony Eight Lin, Fu-Wen Kuo, Yi-Cheng Chou, Hung-Yu Lin and Mei-Chin Lu

Dear Sir,

We would like to express our deep appreciation for your concern regarding the submitted manuscript. We are thankful for your quick and informative reply. We are also grateful for the valuable comments and suggestions from the reviewers aiming to improve the scientific quality of the submitted manuscript. After reading the reviewers’ comments and discussing them with other co-authors, we corrected the manuscript accordingly. The revised and added sentences were highlighted in yellow. We replied to the comments and questions in a point-by-point fashion. Here we enclose the response to the reviewers’ comments.

Response to Reviewer’s Comment 1

We would like to thank the reviewer for her/his comment. We are also grateful for the valuable comments and suggestions from the reviewers aiming to improve the scientific quality of the submitted manuscript.

Reviewer 2 Report

Major comments

It is always difficult to draw conclusion when cell death is concerned, specially when considering cell migration. Lack of cell motility could merely reflect the cell death process, thus to exclude this possibility the authors could for instance use a pancaspase inhibitor to assess cell motility in the presence of their compound.

figure 4 A : Details for FL1 and FL2 staining are lacking in the legend, please explain. Same for panel E

Molecular weights should be shown for WB

Figure 5 : wb analysis of siSQSTM1 need to be shown panel D

Antibodies clone and reference numbers need to be provided in the mat & met section.

minor :

line 126 fig 1B should be Fig 1C

line 149 fig 2C should be Fig 2D etc.. problem with figure N°

Author Response

Reply to the Reviewers’ Comments

Manuscript ID: marinedrugs-850909

Manuscript Title: 13-Acetoxysarcocrassolide Exhibits Cytotoxic Activity Against Oral Cancer Cells Through the Interruption of the Keap1/Nrf2/p62/SQSTM1 Pathway: The Need to Move beyond Classical Concepts

Authors: Yi-Chang Liu, Bo-Rong Peng, Kai-Cheng Hsu, Mohamed El-Shazly, Shou-Ping Shih, Tony Eight Lin, Fu-Wen Kuo, Yi-Cheng Chou, Hung-Yu Lin and Mei-Chin Lu

Dear Sir,

We would like to express our deep appreciation for your concern regarding the submitted manuscript. We are thankful for your quick and informative reply. We are also grateful for the valuable comments and suggestions from the reviewers aiming to improve the scientific quality of the submitted manuscript. After reading the reviewers’ comments and discussing them with other co-authors, we corrected the manuscript accordingly. The revised and added sentences were highlighted in yellow. We replied to the comments and questions in a point-by-point fashion. Here we enclose the response to the reviewers’ comments.

Response to Reviewer’s Comment 2

Comments and Suggestions for Authors

Major comments

  1. It is always difficult to draw conclusion when cell death is concerned, specially when considering cell migration. Lack of cell motility could merely reflect the cell death process, thus to exclude this possibility the authors could for instance use a pancaspase inhibitor to assess cell motility in the presence of their compound.
  • We are thankful for the comment. Since inhibitors cannot affect the action of compounds, we used proteins associated with cell migration (Figure 2C) to further confirm that 13-AC exhibited anti-migration effect.

  1. figure 4 A: Details for FL1 and FL2 staining are lacking in the legend, please explain. Same for panel E
  • We are thankful for the comment. Following the reviewer suggestion, we added necessary information to Figure 4A (FL1 and FL2 staining) legend in the revised manuscript.

  1. Molecular weights should be shown for WB
  • We are thankful for the comment. Following the reviewer suggestion, we added molecular weights in all WB results in the revised manuscript.

  1. Figure 5: wb analysis of siSQSTM1 need to be shown panel D
  • We are thankful for the comment. Following the reviewer suggestion, we added WB analysis of siSQSTM1 to Figure 5D in the revised manuscript.

  1. Antibodies clone and reference numbers need to be provided in the mat & met section.
  • We are thankful for the comment. Following the reviewer suggestion, we added antibodies clone and reference numbers in the mat & met section.

minor :

  1. line 126 fig 1B should be Fig 1C
  • We are thankful for the reviewer comment. Following the reviewer suggestion, we corrected the figures citation in the text in the revised manuscript.

  1. line 149 fig 2C should be Fig 2D etc.. problem with figure N°
  • We are thankful for the reviewer comment. Following the reviewer suggestion, we corrected the figures citation in the text in the revised manuscript.

Reviewer 3 Report

The paper “13-Acetoxysarcocrassolide exhibits cytotoxic activity against oral cancer cells through the interruption of the keap1/Nrf2/p62/SQSTM1 pathway: the need to move beyond classical concepts” by Liu and co-workers investigated the antitumor effect of 13-Acetoxysarcocrassolide against Ca9-22 oral cancer cells and in in vivo xenograft model.

The paper fit within the scope of the journal and present an original topic that is certainly of interest for the readers of Marine Drugs.

Some points need to be improved:

  • In the results section, Authors stated that used different cancer cell lines (Ca9-22, Cal-27, HCT116, Lovo, DLD-1, DU145, LNcap, MCF-7, T47D and 106 Hela) to assess the cytotoxicity of 13-AC, however, in the materials and methods section, the they did not report this information;
  • In my opinion, the Authors should use a non-tumoral cell line to test the cytotoxicity of 13-AC;
  • In figure 1C and 3C, the Authors did not show the results of statistic analysis;
  • Talking about the in vitro study, Authors should replace “dose” with “concentration”;
  • I suggest the Authors revise the English language throughout the article.

Author Response

Reply to the Reviewers’ Comments

Manuscript ID: marinedrugs-850909

Manuscript Title: 13-Acetoxysarcocrassolide Exhibits Cytotoxic Activity Against Oral Cancer Cells Through the Interruption of the Keap1/Nrf2/p62/SQSTM1 Pathway: The Need to Move beyond Classical Concepts

Authors: Yi-Chang Liu, Bo-Rong Peng, Kai-Cheng Hsu, Mohamed El-Shazly, Shou-Ping Shih, Tony Eight Lin, Fu-Wen Kuo, Yi-Cheng Chou, Hung-Yu Lin and Mei-Chin Lu

Dear Sir,

We would like to express our deep appreciation for your concern regarding the submitted manuscript. We are thankful for your quick and informative reply. We are also grateful for the valuable comments and suggestions from the reviewers aiming to improve the scientific quality of the submitted manuscript. After reading the reviewers’ comments and discussing them with other co-authors, we corrected the manuscript accordingly. The revised and added sentences were highlighted in yellow. We replied to the comments and questions in a point-by-point fashion. Here we enclose the response to the reviewers’ comments.

Response to Reviewer’s Comment 3

Comments and Suggestions for Authors

The paper “13-Acetoxysarcocrassolide exhibits cytotoxic activity against oral cancer cells through the interruption of the keap1/Nrf2/p62/SQSTM1 pathway: the need to move beyond classical concepts” by Liu and co-workers investigated the antitumor effect of 13-Acetoxysarcocrassolide against Ca9-22 oral cancer cells and in in vivo xenograft model.

The paper fit within the scope of the journal and present an original topic that is certainly of interest for the readers of Marine Drugs.

Some points need to be improved:

  • In the results section, Authors stated that used different cancer cell lines (Ca9-22, Cal-27, HCT116, Lovo, DLD-1, DU145, LNcap, MCF-7, T47D and 106 Hela) to assess the cytotoxicity of 13-AC, however, in the materials and methods section, the they did not report this information;
  • We are thankful for the comment. Following the reviewer suggestion, we added cell lines information in the materials and methods.

  • In my opinion, the Authors should use a non-tumoral cell line to test the cytotoxicity of 13-AC;
  • We are thankful for the comment. Following the reviewer suggestion, we used HEK293 normal cells for the cytotoxic assay. The results are shown in the following figure. 13-AC showed only a weak toxic effect on normal cells.

  • In figure 1C and 3C, the Authors did not show the results of statistic analysis;
  • We are thankful for the comment. Following the reviewer suggestion, we added the results of the statistical analysis in Figure 1C and Figure 3C in the revised manuscript.

  • Talking about the in vitro study, Authors should replace “dose” with “concentration”;
  • We are thankful for the comment. Following the reviewer suggestion, we changed "dose" to "concentration" in the revised manuscript.

  • I suggest the Authors revise the English language throughout the article.
  • We are thankful for the comment. The manuscript was revised by experts for typos and grammatical mistakes following the reviewer suggestion.

Reviewer 4 Report

The aim of this study was “to evaluate the cytotoxic potential of 13-Acetoxysarcocrassolide (13-AC) against oral cancer cells Ca9-22 along with its mechanism of action using in vitro cellular and in vivo xenograft models”. Although it is known that 13-AC, a marine cytotoxic product isolated from the coral Labophytum crassum, exhibits potent antitumor and immunostimulant effects, the antitumor mechanism of action against oral cancer cells is unclear. Thus, the purpose of the work is original and has been solved with the use of numerous, modern, complementary research techniques.A lot of results are very interesting, both from a cognitive and clinical point of view. The entire text of the paper is clearly written, the results are well presented in majority of the text. However, due to the large number of results, several errors crept into the text, but mainly editorial or related to careless reading of the final version of the work.

I will list them briefly:

  1. Figure 1A citation is missing from the text.
  2. In line 126 – should be cited Figure 1C instead of 1B, please check it and correct.
  3. Was the concentration of 13-AC of 10 µg/mL also checked in wound healing assay, or not? - as compared with this dose used in study of the impact of 13-AC on EMT?
  4. In lines 149-150 – should be cited Figure 2D instead of 2C, and Figure 2E instead of Figure 2D, please check it and correct.
  5. In lines 156-157 in the text – in fact you do not cited the papers concerning gastric cancer [17,20, 21]; please add the appropriate paper, or change the information.
  6. The abbreviation „MMP” should be explained the first time used, g. in line 177 or under Figure 4.
  7. On the 194 line, the NAC doses used to inhibit 13-AC-induced apoptosis are repeated, is this necessary? Why such doses?
  8. Please check figure 4D and 4E carefully, unless these fragments of the figure should be inverted, or correct please the description under the figure. Now on Fig. 4D - is "disruption of MMP", and on Fig. 4E - apoptotic cells (%).
  9. Please check also the Figure 5D carefully, the red bars should be 39% and 32%, it seems that the figure is more (over 40%?). Please correct this.
  10. On line 232, please add caspase 9, and HO-1 and check the description what exactly NAC pretreatment attenuated, and which increased (the description in text is unclear as compared with Figure 5E).
  11. Line 243, 479, 481, and legend to Figure 6 - please unify the 13-AC dose and time (30 days, or 136 days?) administered in vivo.
  12. The Hematoxylin and Eosin (H&E) staining is method essential for recognizing various tissue types and the morphologic changes that form the basis of contemporary cancer diagnosis not for “detecting” changes. Is method to visualize different components of the tissue under a microscope; so I suggest on line 249 – please use “recognized by” instead of “detected by”.
  13. Line 292 – should be after 9-12 h instead of 3 h.
  14. Line 297 – please add reference 44, so should be [43, 44], what about ovary cancer?
  15. Line 304 – ref. 47 – is not paper concerning the leukaemia.
  16. Please check all items cited in this part of the discussion (in red color).
  17. Lines 338-341 vs. 351-352- try to shorten it, because it repeats.
  18. Line 375, please add “HO-1

Small errors:1.      Line 365 – should be “cembranolide” The paper can be accepted for printing after taking into account of my comments above.

Author Response

Reply to the Reviewers’ Comments

Manuscript ID: marinedrugs-850909

Manuscript Title: 13-Acetoxysarcocrassolide Exhibits Cytotoxic Activity Against Oral Cancer Cells Through the Interruption of the Keap1/Nrf2/p62/SQSTM1 Pathway: The Need to Move beyond Classical Concepts

Authors: Yi-Chang Liu, Bo-Rong Peng, Kai-Cheng Hsu, Mohamed El-Shazly, Shou-Ping Shih, Tony Eight Lin, Fu-Wen Kuo, Yi-Cheng Chou, Hung-Yu Lin and Mei-Chin Lu

Dear Sir,

We would like to express our deep appreciation for your concern regarding the submitted manuscript. We are thankful for your quick and informative reply. We are also grateful for the valuable comments and suggestions from the reviewers aiming to improve the scientific quality of the submitted manuscript. After reading the reviewers’ comments and discussing them with other co-authors, we corrected the manuscript accordingly. The revised and added sentences were highlighted in yellow. We replied to the comments and questions in a point-by-point fashion. Here we enclose the response to the reviewers’ comments.

Response to Reviewer’s Comment 4

Comments and Suggestions for Authors

The aim of this study was “to evaluate the cytotoxic potential of 13-Acetoxysarcocrassolide (13-AC) against oral cancer cells Ca9-22 along with its mechanism of action using in vitro cellular and in vivo xenograft models”. Although it is known that 13-AC, a marine cytotoxic product isolated from the coral Labophytum crassum, exhibits potent antitumor and immunostimulant effects, the antitumor mechanism of action against oral cancer cells is unclear. Thus, the purpose of the work is original and has been solved with the use of numerous, modern, complementary research techniques.A lot of results are very interesting, both from a cognitive and clinical point of view. The entire text of the paper is clearly written, the results are well presented in majority of the text. However, due to the large number of results, several errors crept into the text, but mainly editorial or related to careless reading of the final version of the work.

I will list them briefly:

  1. Figure 1A citation is missing from the text.
  • We are thankful for the comment. Following the reviewer suggestion, we added the Figure 1A citation in the revised manuscript.

  1. In line 126 – should be cited Figure 1C instead of 1B, please check it and correct.
  • We are thankful for the reviewer comment. Following the reviewer suggestion, we corrected the figures citation in the text in the revised manuscript.

  1. Was the concentration of 13-AC of 10 µg/mL also checked in wound healing assay, or not? - as compared with this dose used in study of the impact of 13-AC on EMT?
  • We are thankful for the comment. The use of 10 µg/mL of 13-AC demonstrated significant cytotoxic effect so it could not show migration inhibition effect in the wound healing assay.

  1. In lines 149-150 – should be cited Figure 2D instead of 2C, and Figure 2E instead of Figure 2D, please check it and correct.
  • We are thankful for the reviewer comment. Following the reviewer suggestion, we corrected the figures citation in the text in in the revised manuscript.

  1. In lines 156-157 in the text – in fact you do not cited the papers concerning gastric cancer [17,20, 21]; please add the appropriate paper, or change the information.
  • We are thankful for the reviewer comment. Following the reviewer suggestion, we corrected the statement in the revised manuscript.

  1. The abbreviation „MMP” should be explained the first time used, g. in line 177 or under Figure 4.
  • We are thankful for the reviewer comment. Following the reviewer suggestion, we explained the abbreviation „MMP” in line 178.

  1. On the 194 line, the NAC doses used to inhibit 13-AC-induced apoptosis are repeated, is this necessary? Why such doses?
  • We are thankful for the comments. The purpose of this study was to demonstrate that NAC can inhibit apoptosis induced by 13-AC in a dose-dependent manner.

  1. Please check figure 4D and 4E carefully, unless these fragments of the figure should be inverted, or correct please the description under the figure. Now on Fig. 4D - is "disruption of MMP", and on Fig. 4E - apoptotic cells (%).
  • We are thankful for the reviewer comment. Following the reviewer suggestion, we corrected the statements in the revised manuscript.

  1. Please check also the Figure 5D carefully, the red bars should be 39% and 32%, it seems that the figure is more (over 40%?). Please correct this.
  • We are thankful for the reviewer comment. Following the reviewer suggestion, we corrected the statements in the revised manuscript.

  1. On line 232, please add caspase 9, and HO-1 and check the description what exactly NAC pretreatment attenuated, and which increased (the description in text is unclear as compared with Figure 5E).
  • We are thankful for the reviewer comment. Following the reviewer suggestion, we added the description in the revised manuscript.

  1. Line 243, 479, 481, and legend to Figure 6 - please unify the 13-AC dose and time (30 days, or 136 days?) administered in vivo.
  • We are thankful for the reviewer comment. Following the reviewer suggestion, we corrected the statements in the revised manuscript.

  1. The Hematoxylin and Eosin (H&E) staining is method essential for recognizing various tissue types and the morphologic changes that form the basis of contemporary cancer diagnosis not for “detecting” changes. Is method to visualize different components of the tissue under a microscope; so I suggest on line 249 – please use “recognized by” instead of “detected by”.
  • We are thankful for the reviewer comment. Following the reviewer suggestion, we corrected the statement in the revised manuscript.

  1. Line 292 – should be after 9-12 h instead of 3 h.
  • We are thankful for the reviewer comment. Following the reviewer suggestion, we corrected the statement in the revised manuscript.

  1. Line 297 – please add reference 44, so should be [43, 44], what about ovary cancer?
  • In line 297, we discussed the cytotoxic effect of NOV-002 and we cited the corresponding reference [Montero, A.J.; Diaz-Montero, C.M.; Deutsch, Y.E.; Hurley, J.; Koniaris, L.G.; Rumboldt, T.; Yasir, S.; Jorda, M.; Garret-Mayer, E.; Avisar, E., et al. Phase 2 study of neoadjuvant treatment with NOV-002 in combination with doxorubicin and cyclophosphamide followed by docetaxel in patients with HER-2 negative clinical stage II-IIIc breast cancer. Breast Cancer Res Treat 2012, 132, 215-223, doi:10.1007/s10549-011-1889-0.]. In the following line we discussed sulphasalazine and we cited the corresponding reference [Guan, J.; Lo, M.; Dockery, P.; Mahon, S.; Karp, C.M.; Buckley, A.R.; Lam, S.; Gout, P.W.; Wang, Y.Z. The xc- cystine/glutamate antiporter as a potential therapeutic target for small-cell lung cancer: use of sulfasalazine. Cancer Chemother Pharmacol 2009, 64, 463-472, doi:10.1007/s00280-008-0894-4.]
  • Following the reviewer suggestion, we added a new statement “Gambogic acid was found to sensitize ovarian cancer cell line (SKOV-3) to doxorubicin via ROS-mediated apoptosis.” And we cited the corresponding reference.

  1. Line 304 – ref. 47 – is not paper concerning the leukaemia.
  • We are thankful for the reviewer comment. Following the reviewer suggestion, we corrected the statement in the revised manuscript.

  1. Please check all items cited in this part of the discussion (in red color).
  • We are thankful for the reviewer comment. Following the reviewer suggestion, we checked all the cited references corresponding to the test typed in red.

  1. Lines 338-341 vs. 351-352- try to shorten it, because it repeats.
  • We are thankful for the reviewer comment. Following the reviewer suggestion, we removed repetition from the revised manuscript.

  1. Line 375, please add “HO-1”
  • We are thankful for the reviewer comment. Following the reviewer suggestion, we corrected the statement in the revised manuscript.

Small errors:

  1. Line 365 – should be “cembranolide” The paper can be accepted for printing after taking into account of my comments above.

We are thankful for the reviewer comment. Following the reviewer suggestion, we corrected the statement in the revised manuscript.

Round 2

Reviewer 2 Report

Dear Authors,

Many thanks for your revised manuscript. Albeit you have addressed a number of comments successfully, some points still need clarification.

  1. It is always difficult to draw conclusion when cell death is concerned, specially when considering cell migration. Lack of cell motility could merely reflect the cell death process, thus to exclude this possibility the authors could for instance use a pancaspase inhibitor to assess cell motility in the presence of their compound.
  • We are thankful for the comment. Since inhibitors cannot affect the action of compounds, we used proteins associated with cell migration (Figure 2C) to further confirm that 13-AC exhibited anti-migration effect.

You didn't get my point, here. Your compound is indeed likely to inhibit the EMT, but that doesn't address my comment. My point is a if your compound is killing your cells, it is thus not unexpected to find that the cell migration is reduced. Thus should you like to emphasize, like you have done in your manuscript that your compound is able to inhibit cell motility, you need to inhibit apoptosis to demonstrate it. And the best way to do it is to use a pancaspase inhibitor to assess cell migration after stimulation.

  1. figure 4 A: Details for FL1 and FL2 staining are lacking in the legend, please explain. Same for panel E
  • We are thankful for the comment. Following the reviewer suggestion, we added necessary information to Figure 4A (FL1 and FL2 staining) legend in the revised manuscript.

You have change the label panel D in this figure but not panel A

  1. Molecular weights should be shown for WB
  • We are thankful for the comment. Following the reviewer suggestion, we added molecular weights in all WB results in the revised manuscript.

You need not show the theoretical molecular weights, but need to add ticks and molecular sizes on the right or left hand-side of your blot according to the molecular weight ladder  that you must have run side/side with your samples.

Author Response

Reply to the Reviewers’ Comments

Manuscript ID: marinedrugs-850909

Manuscript Title: 13-Acetoxysarcocrassolide Exhibits Cytotoxic Activity Against Oral Cancer Cells Through the Interruption of the Keap1/Nrf2/p62/SQSTM1 Pathway: The Need to Move beyond Classical Concepts

Authors: Yi-Chang Liu, Bo-Rong Peng, Kai-Cheng Hsu, Mohamed El-Shazly, Shou-Ping Shih, Tony Eight Lin, Fu-Wen Kuo, Yi-Cheng Chou, Hung-Yu Lin and Mei-Chin Lu

Dear Sir,

We would like to express our deep appreciation for your concern regarding the submitted manuscript. We are thankful for your quick and informative reply. We are also grateful for the valuable comments and suggestions from the reviewers aiming to improve the scientific quality of the submitted manuscript. After reading the reviewers’ comments and discussing them with other co-authors, we corrected the manuscript accordingly. The revised and added sentences were highlighted using the "Track Changes". We replied to the comments and questions in a point-by-point fashion. Here we enclose the response to the reviewers’ comments.

Response to the Editor’s and Reviewers’ Comments

Many thanks for your revised manuscript. Albeit you have addressed a number of comments successfully, some points still need clarification.

  1. It is always difficult to draw conclusion when cell death is concerned, specially when considering cell migration. Lack of cell motility could merely reflect the cell death process, thus to exclude this possibility the authors could for instance use a pancaspase inhibitor to assess cell motility in the presence of their compound.
  • We are thankful for the comment. Since inhibitors cannot affect the action of compounds, we used proteins associated with cell migration (Figure 2C) to further confirm that 13-AC exhibited anti-migration effect.

You didn't get my point, here. Your compound is indeed likely to inhibit the EMT, but that doesn't address my comment. My point is a if your compound is killing your cells, it is thus not unexpected to find that the cell migration is reduced. Thus should you like to emphasize, like you have done in your manuscript that your compound is able to inhibit cell motility, you need to inhibit apoptosis to demonstrate it. And the best way to do it is to use a pancaspase inhibitor to assess cell migration after stimulation.

  • We are thankful for the comment. The pretreatment with pancaspase inhibitor failed to increase viability and motility of Ca9-22 cells treated with 13-AC in MTT (A) and wound healing assay (B), suggesting that proliferation and migration inhibition by 13-AC is independent on the activation of caspases and these results were different from a previous study in 2014 in which the authors used human gastric carcinoma cells and found that the cells vitality increased by the addition of Z-VAD-FMK (caspase-3 inhibitor) and Z-DEVD-FMK (caspase-9 inhibitor).

A

B

  1. figure 4 A: Details for FL1 and FL2 staining are lacking in the legend, please explain. Same for panel E
  • We are thankful for the comment. Following the reviewer suggestion, we added necessary information to Figure 4A (FL1 and FL2 staining) legend in the revised manuscript.

You have change the label panel D in this figure but not panel A

  • We are thankful for the comment. Following the reviewer suggestion, we modified Figure 4 label panel A description in the revised manuscript.

  1. Molecular weights should be shown for WB
  • We are thankful for the comment. Following the reviewer suggestion, we added molecular weights in all WB results in the revised manuscript.

You need not show the theoretical molecular weights, but need to add ticks and molecular sizes on the right or left hand-side of your blot according to the molecular weight ladder that you must have run side/side with your samples.

  • We are thankful for the comment. Following the reviewer suggestion, we added the molecular weight ladder in all WB results in the revised manuscript.

This manuscript is a resubmission of an earlier submission. The following is a list of the peer review reports and author responses from that submission.

Round 1

Reviewer 1 Report

In this work the authors report on the anticancer effect of a cembrane-type diterpenoid 13-acetoxysarcocrassolide (13-AC) from the soft coral, Sarcophyton crassocaule and Lobophytum crassum and its mechanism of action using in vitro cellular and in vivo xenograft models. The experimental part of the manuscript is devoted to the cytotoxic activity and mechanism of induction of apoptotic cell death connected to this drug. The cytotoxic activity of the drug has been destribed previously on another human cancer cell lines. This work focuses on the effect of 13-AC on human oral cancer cells Ca9-22.

My comments and questions are as follows:

  • „After 72 h of treatment, the IC50 values of 13-AC against Ca9-22, Cal-27, HCT116, Lovo, DLD-1, DU145, LNcap, MCF-7, T47D and Hela cancer cells were 0.94 ± 0.16, 1.31 ± 0.38, 1.36 ± 0.27, 1.38 ± 0.37, 1.64 ± 0.36, 4.85 ± 0.92, 3.93 ± 1.06, 2.44 ± 0.22, 00 ± 0.09 and 4.41 ± 0.75 μg/mL, respectively (Table 1).“ Can the authors compare these values with the cytotoxic activity of cisplatin, which is the drug used for treatment of above mentioned malignacies/cell lines?

  • „The quantification of tetrazolium dye incorporation confirmed that the exposure to a lower dose (1.25 μg/mL) of 13-AC resulted in a significant growth reduction of up to 20%, 60%, and 80% after 24, 48 and 72 h, respectively, compared with the untreated cells.“ Can the authors explain where is the tetrazolium dye incoporated during the MTT assay?

  • „To analyze the effect of 13-AC on the migration of oral cancer cells, Ca9-22 cells were treated with increasing doses (1.25, 2.5 and 5 μg/mL) of 13-AC for 6 and 12 h and the cellular migration was evaluated by wounding healing assay.“ Could the authors compare the results with some control, e.g. cisplatin or a drug conventionaly used to arrest cell migration? Was the antiproliferative activity of tested compound assesed during the wound-healing assay?

  • „We used a comet assay under neutral electrophoresis condition. In a dose-dependent manner, 13-AC (2.5, 5, and 10 μg/mL) promoted DNA fragmentation in cancer cells which was demonstrated by an increase of DNA migration (Figure 2C).“ It is not clearly apparent from Figure 2C (Effect of 13-AC on the migration and DNA damage of human oral cancer Ca9-22 cells) how was the effect of conditions used for this experiment on an untreated control. Can the authors add the results for the negative control according to the treatment period? Can the authors show a magnification used for this testing?

  • „Previous studies showed that the marine cembranoid, 13-AC, induced apoptosis in human gastric carcinoma, hepatocarcinoma, bladder cancer, and nasopharyngeal carcinoma [17,19,20]. Our results showed that 13-AC exhibited antiproliferative and antimigration activities as well as induced DNA damage in Ca9-22 cells (Figures 1 & 2). We then moved further to investigate if 13-AC can induce apoptosis in Ca9-22 cancer cells using flow cytometry and microscopy.“ These findings have already been shown in the previous studies with another cell lines and very likely the findings will be alike also for the human oral cancer cells. It can be helpful to compare results for Ca9-22 with those of the previous testing. Where was 13-AC the best?

  • „As shown in Figure 4B, the treatment with 13-AC (10 μg/mL) for 0.5, 1, 3 and 6 h resulted in 3.32-, 3.96-, 4.98- and 3.05-folds increase in ROS levels, respectively, as compared with the mean fluorescence index (MFI) of the control.“ Can the authors indicate production of which ROS have been increased?

  • How the authors checked the final concentration of their compounds in cell samples?

  • „Since oxidative stress emerged as an attractive target for cancer therapy, we investigated whether apoptosis induction by 13-AC was due to the promotion of ROS generation.“ For which type of cancer therapy are ROS used as a target?

The manuscript includes some inconsistencies e. g:

  • „The cytotoxic activity of 13-AC was determined with MTT assay, flow cytometric analysis, immunofluorescence, immunoprecipitation, Western blotting, and siRNA to identify its molecular mechanism.“
  • „Nuclear factor erythroid 2-related factor 2 (Nrf2) is a transcription factor for cancer hallmarks“
  • „Patients with recurrent or metastatic disease, a remarkable therapeutic advance was achieved through anti-epidermal growth factor receptor (EGFR) therapy and immune checkpoint inhibitors in addition to traditional cytotoxic chemotherapy.“
  • wounding healing assay“
  • flowcytometric assays“

Reviewer 2 Report

The authors stated that  13-AC had potential anticancer agent through the Keap1/Nrf2/p62/SQSTM1 pathway. 

  1. 13-AC induced E-cadherin expression in Fig. 2. Please add the data of vimetin and Slug like as EMT markers. Moreover, please show the molophological chage with EMT in Fig.2.
  2.  The authors should show the results using cal-27 cell to make strong evidence.
  3. In Fig. 2( cell viability), please discuss the result of " for 72h" , why the result did not show dose-dependenet effect unlike 24h and 48h. Ho wever, apoptotic effect showed dose-dependent effect.
  4.  Generally, cancer cells had highly expreseed Nrf2. please disucuss why thir data showed that 13-AC induced Nrf2 expression in spite of its apoptotic effect in Fig. 5.
  5. In Fig. 6, 13-AC showed 100mm3 reduction (from 300 to 200mm3), however, the tumor weight showed big change like as picture. Please discuss about the dissociability of respective data.